# Serological testing of blood donors to characterise the impact of COVID-19 in Melbourne, Australia, 2020

**Dorothy A. Machalek**[1,2]*, **Kaitlyn M. Vette**[3], **Marnie Downes**[4], **John B. Carlin**[4,5], **Suellen Nicholson**[6], **Rena Hirani**[7,8], **David O. Irving**[7,9], **Iain B. Gosbell**[7,10], **Heather F. Gidding**[3,11,12], **Hannah Shilling**[2,4], **Eithandee Aung**[1,2], **Kristine Macartney**[3,10], **John M. Kaldor**[1]

1 The Kirby Institute, University of New South Wales, Sydney, Australia, 2 Centre for Women's Infectious Diseases, The Royal Women's Hospital, Melbourne, Australia, 3 National Centre for Immunisation Research and Surveillance, Sydney, Australia, 4 Murdoch Children's Research Institute, Melbourne, Australia, 5 Department of Paediatrics and School of Population and Global Health, University of Melbourne, Melbourne, Australia, 6 Victorian Infectious Diseases Reference Laboratory, The Royal Melbourne Hospital, The Peter Doherty Institute for Infection and Immunity, Melbourne, Australia, 7 Clinical Services and Research, Australian Red Cross Lifeblood, Sydney, Australia, 8 Department of Molecular Sciences, Macquarie University, Sydney, Australia, 9 Faculty of Health, University of Technology Sydney, Sydney, Australia, 10 School of Medicine, Western Sydney University, Sydney, Australia, 11 Faculty of Medicine and Health, The University of Sydney Northern Clinical School, Sydney, Australia, 12 Women and Babies Research, Kolling Institute, Northern Sydney Local Health District, Sydney, Australia

* dmachalek@kirby.unsw.edu.au

**Data Availability Statement:** All relevant data are within the manuscript and its Supporting Information files.

## Abstract

Rapidly identifying and isolating people with acute SARS-CoV-2 infection has been a core strategy to contain COVID-19 in Australia, but a proportion of infections go undetected. We estimated SARS-CoV-2 specific antibody prevalence (seroprevalence) among blood donors in metropolitan Melbourne following a COVID-19 outbreak in the city between June and September 2020. The aim was to determine the extent of infection spread and whether sero-prevalence varied demographically in proportion to reported cases of infection. The design involved stratified sampling of residual specimens from blood donors (aged 20–69 years) in three postcode groups defined by low (<3 cases/1,000 population), medium (3–7 cases/1,000 population) and high (>7 cases/1,000 population) COVID-19 incidence based on case notification data. All specimens were tested using the Wantai SARS-CoV-2 total antibody assay. Seroprevalence was estimated with adjustment for test sensitivity and specificity for the Melbourne metropolitan blood donor and residential populations, using multilevel regression and poststratification. Overall, 4,799 specimens were collected between 23 November and 17 December 2020. Seroprevalence for blood donors was 0.87% (90% credible interval: 0.25–1.49%). The highest estimates, of 1.13% (0.25–2.15%) and 1.11% (0.28–1.95%), respectively, were observed among donors living in the lowest socioeconomic areas (Quintiles 1 and 2) and lowest at 0.69% (0.14–1.39%) among donors living in the highest socioeconomic areas (Quintile 5). When extrapolated to the Melbourne residential population, overall seroprevalence was 0.90% (0.26–1.51%), with estimates by demography groups similar to those for the blood donors. The results suggest a lack of extensive community transmission and good COVID-19 case ascertainment based on routine testing during

**Funding:** The study was supported by the Victorian Government Department of Health, and Snow Medical Foundation (CT28701/G207593). The Australian Government funds Australian Red Cross Lifeblood to provide blood, blood products and services to the Australian community. The funders have no role in study design, data collection and analysis, decision to publish, or preparation of the manuscript.

**Competing interests:** The authors have declared that no competing interests exist.

Victoria's second epidemic wave. Residual blood donor samples provide a practical epidemiological tool for estimating seroprevalence and information on population patterns of infection, against which the effectiveness of ongoing responses to the pandemic can be assessed.

## Introduction

In 2020, Australia experienced two distinct COVID-19 epidemic waves of the original variant of the SARS-CoV-2 virus initially detected in Wuhan, China, in December 2019. The first occurred between March and April across the country [1]. Control measures introduced throughout March included the closure of Australian borders and enforceable stay-at-home directives. By the end of April, Australia had successfully suppressed SARS-CoV-2 transmission with a cumulative 7,345 infections, most in returned travellers and their primary contacts. Restrictions began easing in early May [1]. However, Victoria experienced a resurgence in infections in mid-June, with 18,454 cases notified between 14 June and 30 September. In contrast to the first wave, virtually all COVID-19 cases detected were locally acquired, with the vast majority in residents of the state capital, Melbourne [1,2]. Extensive movement restrictions and other public health and social measures led to a sustained decline in incident cases to zero by November and a gradual easing of restrictions [1].

Rapid identification and isolation of people with acute SARS-CoV-2 infection via nucleic acid testing (NAT) is a core strategy for containing COVID-19 but is likely to miss a proportion of infections, particularly in people with few or no symptoms or those who do not access testing [3,4]. Furthermore, case detection levels may vary by demographic characteristics and by health service utilisation. Population surveys of SARS-CoV-2 antibody prevalence (serosurveys) can provide a better understanding of the cumulative prevalence of past infection and associated patterns. Surveys conducted in mid-2020 after Australia's first wave used residual blood specimens from blood donors, pregnant women and people undergoing outpatient pathology testing [5,6]. They found very low infection levels during the first COVID-19 epidemic wave, with the three populations providing similar results. While these results support the observation that community transmission was low during the first COVID-19 epidemic wave, there were too few positive cases to draw conclusions about the extent to which serological patterns of infection matched those apparent in notified cases.

Given the substantial number of notified cases arising in Victoria's second wave, we surveyed Melbourne blood donors using a stratified sampling method informed by COVID-19 notification rates. The aim was to estimate the prevalence of SARS-CoV-2 specific antibodies in this population and determine whether it varied demographically in proportion to reported cases of infection in Melbourne.

## Materials and methods

Procedures followed those established for Australia's first national COVID-19 serosurvey [5]. In Victoria, all blood donations are processed by Australian Red Cross Lifeblood (Lifeblood) through a single processing centre in Melbourne. Demographic information available for each donor included: sex (female, male), age group (20–29, 30–39, 40–49, 50–59, 60–69 years) and postcode of residence. The design involved stratified sampling of residual specimens provided by blood donors from three groups of Melbourne metropolitan postcodes of residence defined by COVID-19 case notification data from the start of the pandemic to 28 October 2020 [7].

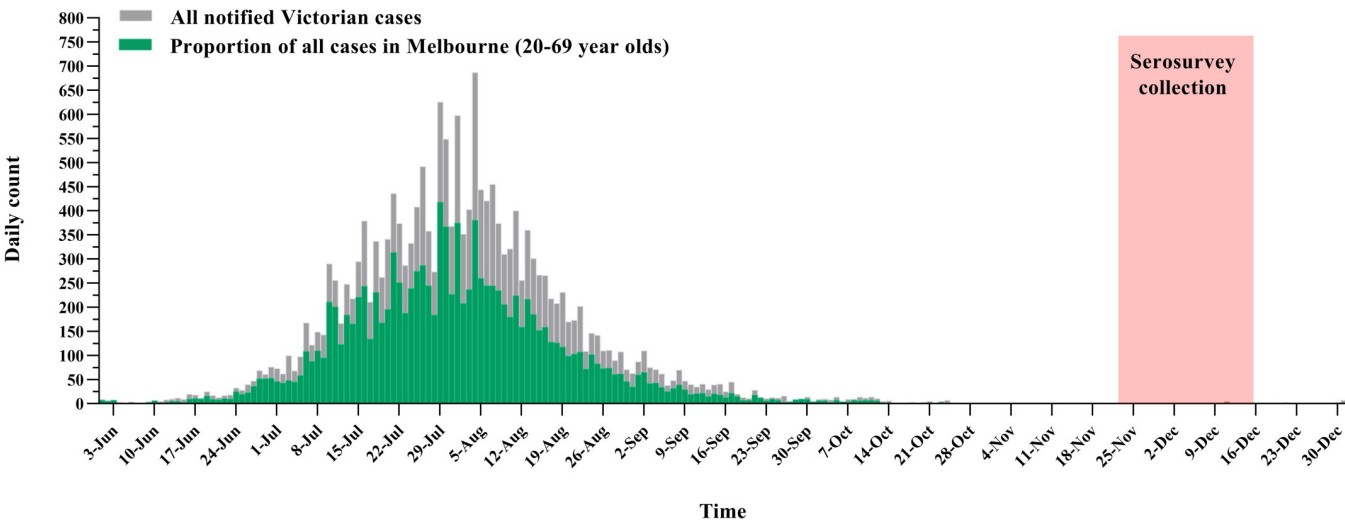

**Fig 1. Count of COVID-19 case notifications between 1 June and 31 December 2020 in Victoria overall (grey bars) and among residents of metropolitan Melbourne aged 20–69 years (green bars), and timing of specimen collection.**

The groups were classified as: low (fewer than 3 cases per 1,000 residents; 50% of metropolitan Melbourne postcodes); medium (3–7 cases per 1,000; 30% of postcodes); and high (more than 7 cases per 1,000; 20% of postcodes). The sample included almost a third of postcodes (n = 35/110) in the low incidence group, half those (n = 27/59) in the medium incidence group, and all (n = 37) in the high incidence group (S1 Table). Postcodes for inclusion in the low and medium incidence groups were randomly selected, considering available donor numbers and the feasibility of collecting 1,600 sequential specimens in each group over four weeks. Within each postcode group, consecutive eligible specimens were then collected. The collection took place between 23 November to 17 December 2020, approximately four months after Victoria's peak in daily notifications and prior to the introduction of COVID-19 vaccines (Fig 1).

Specimens were tested at the Victorian Infectious Disease Reference Laboratory (Melbourne, Australia), using the Wantai SARS-CoV-2 total antibody enzyme-linked immunosorbent assay (ELISA; Beijing Wantai Biological Pharmacy Enterprise Co Ltd, China). We selected this assay based on in-house validations [5], including head-to-head comparisons of available commercial tests [8]. As previously described [5], we assessed Wantai performance on 102 stored specimens that were confirmed RT-PCR positive and collected more than 14 days post-symptom onset (median = 31 days, IQR = 21–40, max = 130). A positive result was found in 97, giving a test sensitivity of 97/102 = 95.1% (95% confidence interval: 88.9–98.4). We tested 800 (pre-pandemic) blood donor specimens from May 2019 and found three positive, giving a test specificity of 797/800 = 99.6% (98.9–99.9) [5].

We reported crude seropositivity and estimated seroprevalence overall, by sampling stratum, and demographic subcategories (sex, age group, socioeconomic quintiles). Quintiles of socioeconomic disadvantage (lowest to highest; from here on referred to as 'Socioeconomic quintiles') were assigned to postcodes based on the Australian Bureau of Statistics 2016 Index of Relative Socio-economic Disadvantage Ranking within Victoria [9]. Seroprevalence was estimated using Bayesian methods to adjust for sensitivity and specificity, incorporating the statistical uncertainty in these estimated values [10]. We applied multilevel regression and poststratification to model the variation in seroprevalence by postcode, sampling stratum, sex, age group and socioeconomic status. A weighted population prevalence estimate based on the population distribution across all possible combinations of these covariates was obtained for

each of the Melbourne blood donor and resident populations aged 20–69 years (S1 File). Estimation assumed a uniform prior distribution for seroprevalence (generally preferred on the basis of being less subjective). In a sensitivity analysis, we used an alternative (more realistic) prior distribution, which focused on values for seroprevalence below 5%. We summarised seroprevalence estimates using the median (point estimate) and 90% credible interval (CrI) of the corresponding posterior probability distribution. Analyses were performed in R using the rstan package.

Cumulative COVID-19 notifications for Melbourne residents aged 20–69 years from the start of the pandemic to 21 November 2020 (14 days before the median specimen collection date) were calculated by sampling stratum and demographic subcategories using anonymised notification data supplied by the Victorian Department of Health. Rates were expressed per 100,000 estimated resident population [11]. For calculation of the infection-to-case ratio, we multiplied the estimated seroprevalence (with 90% CrI) for the Melbourne metropolitan population aged 20–69 years by the estimated size of the population to calculate the total number of people infected (reported per 100,000 population). This estimate was compared with the cumulative number of notified COVID-19 cases reported in the same age group from the start of the pandemic to 14 days before the median specimen collection date.

Ethics approvals were granted by the Sydney Children's Hospital Network Human Research Ethics Committee (HREC/17/SCHN/245) and Lifeblood HREC (2020#07). A waiver of individual consent was granted to use residual and de-identified samples.

## Results

Overall, 4799 specimens were collected: 1600, 1600 and 1599 in each of the low, medium, and high incidence strata, respectively. The median specimen collection date was 5 December 2020, with no difference by postcode sampling stratum (Fig 1). Nearly two-thirds of donors in the sample (73.5%) were under 50 years, and 48.9% lived in the highest socioeconomic areas (Quintiles 4 or 5) (Table 1). The distribution of available demographic characteristics of the study population was similar to that for the broader Melbourne blood donor and metropolitan populations (Table 1). However, there were notable differences in the distribution of blood donors within each sampling stratum by socioeconomic areas. For example, a greater proportion of donors from low incidence postcodes lived in higher socioeconomic areas (78.9% Quintiles 4 or 5 versus 11.3% Quintiles 1 or 2). Conversely, among donors from high incidence postcodes, a greater proportion lived in lower socioeconomic areas (65.6% Quintiles 1 or 2 versus 11.5% Quintiles 4 or 5 (Table 1).

Overall, 77 (1.60%) blood donors had detectable SARS-CoV-2 antibodies: 20 (1.25%) in the low incidence, 29 (1.81%) in the medium incidence, and 28 (1.75%) in the high incidence strata (Table 2). Estimated seroprevalence for metropolitan Melbourne blood donors aged 20–69 years was 0.87% (0.25–1.49%): 0.73% (0.17–1.40%) for donors living in low incidence postcodes; 0.97% (0.25–1.73%) in medium incidence postcodes; and 1.06% (0.27–1.82%) in high incidence postcodes. There was a suggestion of a U-shaped relationship between seroprevalence and age, with higher point estimates of 0.94% (0.24–1.72) at age 20–29 years, declining to 0.73% (0.17–1.41%) at age 40–49 years, then increasing to 0.86% (0.20–1.71%) at age 60–69 years. The highest seroprevalence estimates, of 1.13% (0.25–2.15%) and 1.11% (0.28–1.95%), respectively, were observed among donors living in the lowest socioeconomic areas (Quintiles 1 and 2). Seroprevalence was lowest at 0.69% (0.14–1.39%) among donors living in the highest socioeconomic areas (Quintile 5) (Fig 2 and Table 2). When extrapolated to the metropolitan Melbourne residential population, overall seroprevalence was 0.90% (0.26–1.51%), with

**Table 1. Demographic characteristics of the study populations, by sampling stratum compared to Melbourne blood donors and residential populations aged 20–69 years.**

| Variable | Overall sample | Low incidence[a] | Medium incidence[a] | High incidence[a] | Melbourne blood donor population[b] | Melbourne resident population[c] |
|---|---|---|---|---|---|---|
| | 4,799 | 1,600 | 1,600 | 1,599 | 29,731 | 2,678,532 |
| **Sex** | | | | | | |
| Female | 2,400 (50.0) | 800 (50.0) | 800 (50.0) | 800 (50.0) | 14,202 (47.8) | 1,365,140 (51.0) |
| Male | 2,399 (50.0) | 800 (50.0) | 800 (50.0) | 799 (50.0) | 15,529 (52.2) | 1,313,392 (49.0) |
| **Age group** | | | | | | |
| 20–29 years | 1,302 (27.1) | 401 (25.1) | 451 (28.2) | 450 (28.1) | 9,400 (31.6) | 643,911 (24.0) |
| 30–39 years | 1,309 (27.3) | 338 (21.1) | 462 (28.9) | 509 (31.8) | 7,950 (26.7) | 636,218 (23.8) |
| 40–49 years | 916 (19.1) | 295 (18.4) | 307 (19.2) | 314 (19.6) | 5,308 (17.9) | 556,067 (20.8) |
| 50–59 years | 760 (15.8) | 333 (20.8) | 225 (14.1) | 202 (12.6) | 4,458 (15.0) | 476,586 (17.8) |
| 60–69 years | 512 (10.7) | 233 (14.6) | 155 (9.7) | 124 (7.8) | 2,615 (8.8) | 365,750 (13.7) |
| **Socioeconomic quintiles[d]** | | | | | | |
| Quintile 1 (lowest) | 760 (15.8) | 0 (0.0) | 180 (11.3) | 580 (36.3) | 2,924 (9.8) | 426,472 (15.9) |
| Quintile 2 | 923 (19.2) | 180 (11.3) | 275 (17.2) | 468 (29.3) | 3,221 (10.8) | 317,970 (11.9) |
| Quintile 3 | 752 (15.7) | 157 (9.8) | 247 (15.4) | 348 (21.8) | 5,222 (17.6) | 493,540 (18.4) |
| Quintile 4 | 748 (15.6) | 334 (20.9) | 288 (18.0) | 126 (7.8) | 7,494 (25.2) | 586,731 (21.9) |
| Quintile 5 (highest) | 1597 (33.3) | 928 (58.0) | 610 (38.1) | 59 (3.7) | 1,0870 (36.6) | 853,819 (31.9) |
| Missing[e] | 19 (0.4) | 1 (0.1) | 0 (0.0) | 18 (1.1) | – | – |

[a]Sampling strata were defined by COVID-19 case notification data to 28 October 2020: <3 cases/1,000 population (Low incidence postcodes); 3–7 cases/1,000 population (Medium incidence postcodes); >7 cases/1,000 population (High incidence postcodes) (S1 Table).

[b]Estimates based on counts of Lifeblood plasma donors in the 2019 calendar year for the included postcode groups (internal communications).

[c]Estimates based on counts of persons place of usual residence from the ABS 2016 Census for the relevant postcodes.

[d]Socioeconomic status was assigned from residential postcode based on ABS 2016 Index of relative socioeconomic disadvantage ranking within Victoria [11].

[e]One postcode did not have an index score.

estimates by sampling stratum, sex, age-group, and socioeconomic status very similar to those for the blood donors (Table 2).

The cumulative number of SARS-CoV-2 infections was 900 per 100,000 population, based on seroprevalence compared with 480 cases per 100,000 population notified to 21 November 2020. This gave an infection-to-case ratio of 1.9, with an upper 90% credible interval limit of 3.1.

Although the credible intervals for seroprevalence estimates by demographic groups largely overlapped, seroprevalence estimates were broadly consistent with corresponding patterns observed for notified cases. This was evident by sampling stratum where cumulative case notification rates were 198 per 100,000 in low incidence postcodes compared with 1,160 per 100,000 population in high incidence postcodes. Similarly, cumulative notification rates were highest, at 892 per 100,000, in the lowest socioeconomic areas (Quintile 1) and lowest, at 263 per 100,000, in the highest socioeconomic areas (Quintile 5). A consistent pattern was not seen for age where cumulative notification rates continued to decline with increasing age group (Fig 3).

## Discussion

In this study, we evaluated 4,799 blood donors for SARS-CoV-2 antibodies to investigate the extent to which infection occurred in metropolitan Melbourne following Victoria's second

**Table 2. Crude and estimated SARS-CoV-2 seroprevalence and 90% credible intervals (CrI) for the Melbourne blood donor population (A) and metropolitan Melbourne resident population (B) aged 20–69 years.**

| | Crude estimates N (%) | Melbourne blood donor population | | Melbourne resident population | |
|---|---|---|---|---|---|
| | | Primary analysis[a] % (90% CrI) | Sensitivity analysis[b] % (90% CrI) | Primary analysis[a] % (90% CrI) | Sensitivity analysis[b] % (90% CrI) |
| **Overall population** | 77 (1.60) | 0.87 (0.25–1.49) | 0.79 (0.20–1.43) | 0.90 (0.26–1.51) | 0.82 (0.21–1.46) |
| **Sampling stratum[c]** | | | | | |
| Low incidence | 20 (1.25) | 0.73 (0.17–1.40) | 0.66 (0.13–1.33) | 0.73 (0.17–1.38) | 0.65 (0.13–1.32) |
| Medium incidence | 29 (1.81) | 0.97 (0.25–1.73) | 0.88 (0.19–1.68) | 0.99 (0.26–1.77) | 0.91 (0.20–1.71) |
| High incidence | 28 (1.75) | 1.06 (0.27–1.82) | 0.98 (0.20–1.76) | 1.06 (0.27–1.85) | 0.98 (0.20–1.79) |
| **Sex** | | | | | |
| Female | 35 (1.46) | 0.78 (0.22–1.42) | 0.71 (0.17–1.36) | 0.80 (0.23–1.44) | 0.73 (0.18–1.38) |
| Male | 42 (1.75) | 0.94 (0.25–1.68) | 0.85 (0.19–1.59) | 0.98 (0.26–1.72) | 0.89 (0.20–1.64) |
| **Age group** | | | | | |
| 20–29 years | 25 (1.92) | 0.94 (0.24–1.72) | 0.86 (0.19–1.64) | 1.00 (0.26–1.79) | 0.91 (0.20–1.72) |
| 30–39 years | 21 (1.60) | 0.88 (0.22–1.58) | 0.80 (0.18–1.52) | 0.89 (0.23–1.59) | 0.82 (0.18–1.53) |
| 40–49 years | 10 (1.09) | 0.73 (0.17–1.41) | 0.66 (0.14–1.36) | 0.75 (0.17–1.42) | 0.67 (0.14–1.37) |
| 50–59 years | 11 (1.45) | 0.79 (0.18–1.50) | 0.72 (0.15–1.43) | 0.84 (0.20–1.55) | 0.76 (0.16–1.48) |
| 60–69 years | 10 (1.95) | 0.86 (0.20–1.71) | 0.78 (0.16–1.64) | 0.91 (0.22–1.76) | 0.83 (0.17–1.69) |
| **Socioeconomic quintiles[d]** | | | | | |
| Quintile 1 (lowest) | 13 (1.71) | 1.13 (0.25–2.15) | 1.04 (0.18–2.09) | 1.11 (0.25–2.10) | 1.03 (0.19–2.04) |
| Quintile 2 | 18 (1.95) | 1.11 (0.28–1.95) | 1.02 (0.20–1.90) | 1.10 (0.28–1.94) | 1.02 (0.20–1.90) |
| Quintile 3 | 14 (1.86) | 0.98 (0.24–1.84) | 0.90 (0.18–1.77) | 0.99 (0.24–1.85) | 0.91 (0.18–1.78) |
| Quintile 4 | 10 (1.34) | 0.77 (0.17–1.49) | 0.69 (0.13–1.43) | 0.75 (0.17–1.46) | 0.68 (0.13–1.41) |
| Quintile 5 (highest) | 22 (1.38) | 0.69 (0.14–1.39) | 0.62 (0.11–1.32) | 0.68 (0.14–1.37) | 0.61 (0.11–1.30) |

[a]Estimation assumed a uniform prior distribution for seroprevalence.

[b]Estimation assumed an alternative prior distribution, which focused on values for seroprevalence below 5%.

[c]Low incidence postcodes: <3 cases/1,000 population; Medium incidence postcodes: 3–7 cases/1,000 population; High incidence postcodes: >7 cases/1,000 population (S1 Table).

[d]Assigned from residential postcode based on ABS 2016 Index of relative socioeconomic disadvantage ranking within Victoria.

epidemic wave in 2020. We found that overall SARS-CoV-2 seroprevalence in donors was 0.9%, with an upper 90% credible interval of 1.5%. Seroprevalence by sampling stratum, sex, and socioeconomic status in donors was broadly consistent with corresponding patterns observed for notified cases for metropolitan Melbourne, but the patterns were not as pronounced as the patterns observed for case notifications. When extrapolated to the Melbourne metropolitan population, the infection-to-case ratio was 1.9 with an upper 90% credible interval of 3.1.

The results suggest a lack of extensive community transmission and good COVID-19 case ascertainment based on routine testing during the second (and at the time Australia's largest) COVID-19 epidemic wave. Testing was widely available and strongly encouraged, including for people with the mildest of symptoms. Testing rates increased over 3-fold in the weeks leading up to the peak of the second wave, from 1,800 tests per 100,000 population over a two-week reporting period in June to over 5,000 tests per 100,000 population in late July and early August, with test positivity peaking at 1.7% in the first two weeks in August [12]. The proportion of cases with an unknown source peaked at nearly 60% before falling to fewer than 10% by the end of September, as case numbers declined, and contact tracing was more effective.

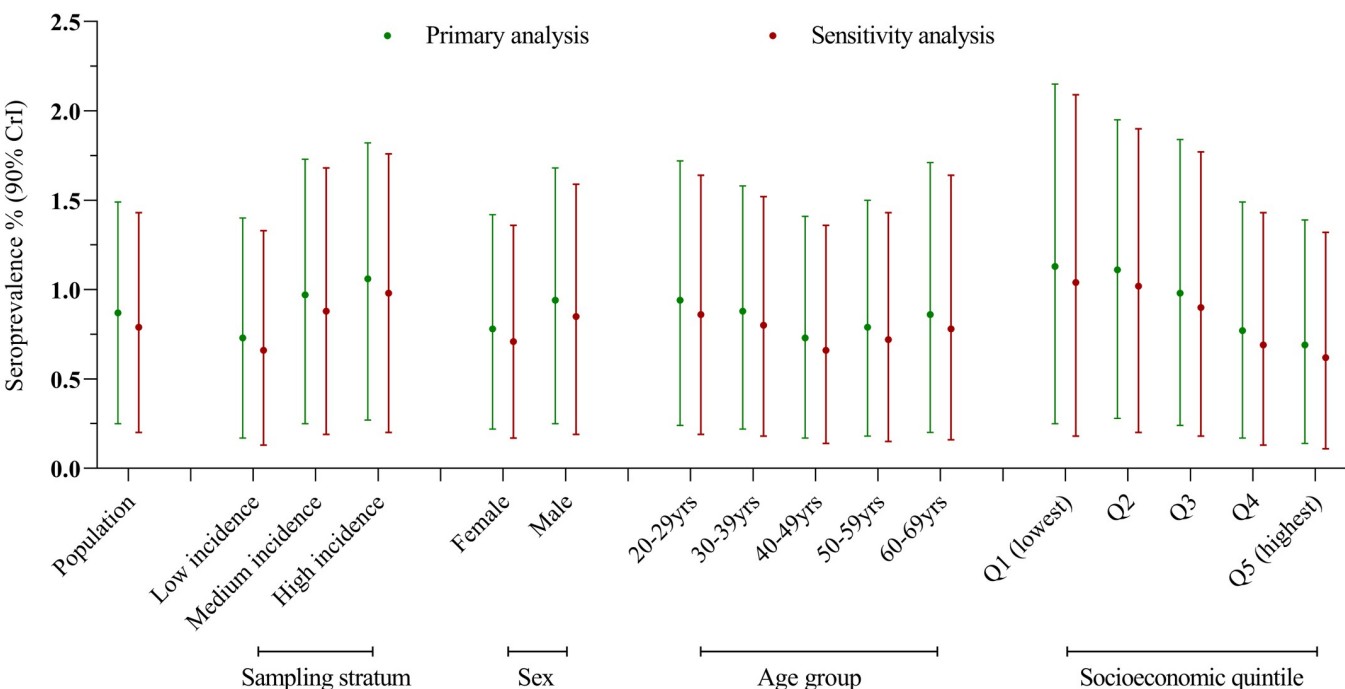

**Fig 2. Estimated SARS-CoV-2 seroprevalence and 90% credible intervals (CrI) for metropolitan Melbourne blood donors aged 20–69 years.** Estimation in the primary analysis assumed a uniform prior distribution for seroprevalence. Estimation in the sensitivity analysis assumed an alternative prior distribution, which focused on values for seroprevalence below 5%. Sampling strata were defined by COVID-19 case notification data to 28 October 2020: <3 cases/1,000 population (Low incidence postcodes); 3–7 cases/1,000 population (Medium incidence postcodes); >7 cases/1,000 population (High incidence postcodes) (S1 Table). Socioeconomic status was assigned from residential postcode based on ABS 2016 Index of relative socioeconomic disadvantage ranking within Victoria [9].

A key methodological challenge for surveillance of any type is sampling populations of interest in a manner that is broadly representative of the underlying target population. Blood donor specimens have been used to monitor the prevalence of antibodies to a wide range of infectious agents, and for COVID-19, have been adopted by serosurveillance programs in the USA, UK and elsewhere to infer the spread of population infection [13,14]. However, it is also well recognised that blood donors are a selected population, a limitation of using these samples. Blood donors tend to be healthier, may generally have a higher average income and education, and may also be at lower risk of COVID-19 infection than the general population [15,16]. To help address this bias, we collected residual specimens using a sampling approach that stratified based on case notification data to provide broad representation across the metropolitan Melbourne populations at risk of COVID-19. While our seroprevalence estimates were broadly consistent with corresponding relationships observed for notified cases, the patterns were not as pronounced. Of note, the ratio between the high and low incidence strata was 5.9-fold based on notified cases but only 1.5-fold based on seroprevalence. Similarly, the ratio between the lowest and highest socioeconomic quintiles was 3.4-fold based on notified cases, but only 1.6-fold based on seroprevalence. A previous study that used routine notification data to inform sampling to estimate seroprevalence in a single urban area in Houston, Texas, found a much greater divergence in seroprevalence between areas with high (18%) and low (10%) case notifications and between demographic groups known to be disproportionately affected by the pandemic. Overall seroprevalence in the city was 14%, suggesting extensive community transmission. The Houston study employed a random sampling approach of the general population to recruit consenting participants for serological assessment. This

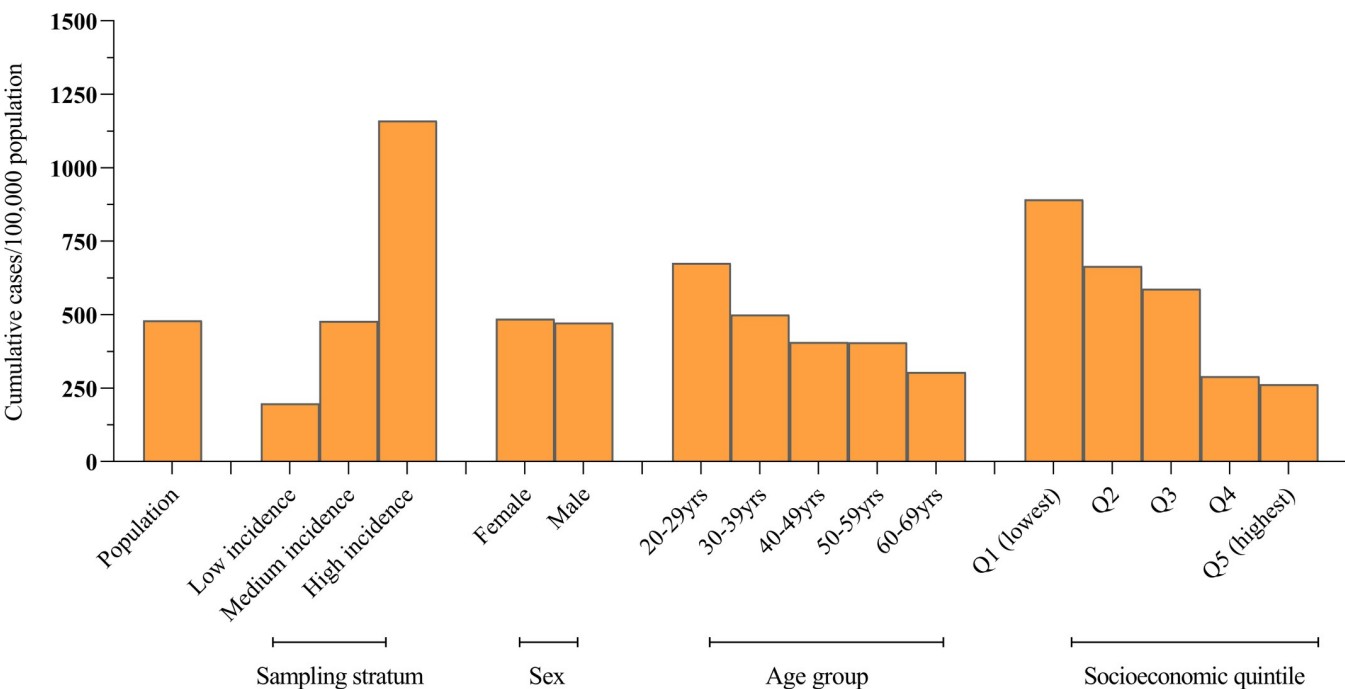

**Fig 3. Cumulative COVID-19 notifications for Melbourne residents aged 20–69 years from the start of the pandemic to 21 November 2020, by sampling stratum and demographic characteristics.** Sampling strata were defined by COVID-19 case notification data to 28 October 2020: <3 cases/1,000 population (Low incidence postcodes); 3–7 cases/1,000 population (Medium incidence postcodes); >7 cases/1,000 population (High incidence postcodes) (S1 Table). Socioeconomic status was assigned from residential postcode based on ABS 2016 Index of relative socioeconomic disadvantage ranking within Victoria [11].

difference in the study design and the higher and more geographically uniform COVID-19 incidence may have led to a greater ability to identify differences.

Taken together, these data highlight the potential difficulty of estimating seroprevalence using blood donor sampling in a setting of the relatively low incidence of COVID-19 infection that is likely to be highly clustered within particular subgroups. Based on notifications, the Melbourne outbreak disproportionately affected those living in areas of socioeconomic disadvantage, who were at higher risk due to occupational and domestic factors [17]. This population may have less overlap with the blood donor population available for sampling. Our estimates of seroprevalence in Melbourne blood donors and in the general population corrected for differences between our sample and these target populations with respect to postcode, sex, and age group. However, we could not account for potentially important predictors of infection risk such as occupation and cultural and social categories (country of birth, language spoken at home, household density), which may differ between blood donors and the general population. Furthermore, we were only able to use an area-based measure of socioeconomic status while recognising that an individual's characteristics may not match those of their area of residence [18]. In the absence of additional individual-level data to include in the modelling, our results may have been biased towards lower overall seroprevalence estimates.

Additional methodological considerations when interpreting the results of this study include the following. First, the estimate of the sensitivity of the Wantai total antibody assay was derived from specimens obtained from NAT-positive people diagnosed early in the pandemic, when testing was likely to target people with more severe symptoms [8,19]. The sensitivity of the test in people who experienced mild illness or were asymptomatic at the time of their infection may be lower, resulting in a downward bias in seroprevalence estimates [20].

Furthermore, the credible intervals of our seroprevalence point estimates largely overlapped. Finally, we could not distinguish between infections that occurred in the first wave from those in the second wave. However, the likely contribution from the first wave would be minimal since most cases in Melbourne occurred in the second outbreak.

Despite the above, seroprevalence estimates based on blood donor sampling can provide valuable information on population patterns of infection, against which the effectiveness of ongoing responses to the pandemic can be assessed [14]. Surveillance studies conducted in settings that have experienced widespread COVID-19 transmission, have found similar seroprevalence estimates among blood donors and household surveys targeting the general population [21]. Furthermore, blood donors are a well-defined healthy, and demographically diverse population with respect to age, sex, and geography, with specimen collection and storage systems well embedded into routine workloads. This provides a practical mechanism for sampling that is convenient and repeatable over time to produce comparable estimates [14]. In the UK and USA, where surveys among blood donor populations have been used to track changes in infection rates, steady increases in seroprevalence over time have been reported, consistent with the reported trends in COVID-19 infection occurrence in these countries [14,22–24].

In conclusion, the overall low seroprevalence estimates reported in this study suggest limited community transmission during Melbourne's surge in cases in mid-2020 and reflect the successful impacts of control measures including widespread availability of testing, extensive contact tracing and social distancing measures. Furthermore, while serosurveillance of residual blood donor samples may underestimate infection spread in certain settings or pockets of localised transmission (particularly in very low prevalence settings), well-designed serosurveillance studies using blood donor populations can contribute important data on both the extent that the virus had spread in the community, and the impact of mitigating strategies which now include vaccination. Australia's COVID-19 vaccination program commenced in February 2021 and by the end of that year had achieved some of the highest vaccination rates in the world, with a third dose being rolled out to people who received their primary course at least 3 to 4 months prior [25]. However, the COVID-19 pandemic continues to change with the outbreak in June 2021 of the Delta variant and the emergence in November 2021 of the highly transmissible Omicron strain. As Australia scales back public health and social measures, blood donors serosurveys can be used to track both the spread of infection as well as levels of vaccine-induced immunity in the population.

## Supporting information

**S1 Table. List of postcodes, by sampling group.**
(DOCX)

**S1 File. Multilevel regression and poststratification modelling.**
(DOCX)

**S2 File.**
(XLSX)

## Acknowledgments

We would like to thank Elizabeth Knight and Jesse Fryk from Australian Red Cross Lifeblood for their assistance in study implementation and specimen collection, respectively. We also thank Daniel Sjmons, Rianne Brizuela and Thomas Holgate for the VIDRL Serology laboratory for testing support. Finally, we thank the Victorian Department of Health for their advice

and input, particularly Catherine Streeton, Sarah Yallop, Amelia Matlock, Aneesha Heranjal, Philimon Haile and Daniel Kidd.

## Author Contributions

**Conceptualization:** Dorothy A. Machalek, Kaitlyn M. Vette, John B. Carlin, Suellen Nicholson, Rena Hirani, David O. Irving, Iain B. Gosbell, Heather F. Gidding, Kristine Macartney, John M. Kaldor.

**Data curation:** Dorothy A. Machalek, Suellen Nicholson, Rena Hirani, David O. Irving, Iain B. Gosbell.

**Formal analysis:** Dorothy A. Machalek, Marnie Downes, John B. Carlin.

**Funding acquisition:** Dorothy A. Machalek, John B. Carlin, Kristine Macartney, John M. Kaldor.

**Investigation:** John B. Carlin.

**Methodology:** Dorothy A. Machalek, Kaitlyn M. Vette, Marnie Downes, John B. Carlin, Suellen Nicholson, Rena Hirani, David O. Irving, Iain B. Gosbell, Heather F. Gidding, Kristine Macartney, John M. Kaldor.

**Project administration:** Dorothy A. Machalek, Kaitlyn M. Vette, Rena Hirani, Hannah Shilling, Eithandee Aung.

**Validation:** Marnie Downes.

**Writing – original draft:** Dorothy A. Machalek.

**Writing – review & editing:** Dorothy A. Machalek, Kaitlyn M. Vette, Marnie Downes, John B. Carlin, Suellen Nicholson, Rena Hirani, David O. Irving, Iain B. Gosbell, Heather F. Gidding, Hannah Shilling, Eithandee Aung, Kristine Macartney, John M. Kaldor.

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
