## [Decision Letter · Decision Letter 0]

18 May 2022

PONE-D-22-06787SEROLOGICAL TESTING OF BLOOD DONORS TO CHARACTERISE THE IMPACT OF COVID-19 IN MELBOURNE, AUSTRALIA, 2020PLOS ONE

Dear Dr. Dorothy Machalek, 

Thank you for submitting your manuscript to PLOS ONE. After careful consideration, we feel that it has merit but does not fully meet PLOS ONE’s publication criteria as it currently stands. Therefore, we invite you to submit a revised version of the manuscript that addresses the points raised during the review process. The manuscript needs revisions recommended by the reviewers.   

We look forward to receiving your revised manuscript.

Kind regards,

Shawky M Aboelhadid, PhD

Academic Editor

PLOS ONE

Journal Requirements:

“The study was supported by the Victorian Government Department of Health and Snow Medical Foundation (CT28701/G207593). The Australian government funds Australian Red Cross Lifeblood to provide blood, blood products and services to the Australian community.”

“The funders have no role in study design, data collection and analysis, decision to publish, or preparation of the manuscript.”

Reviewers' comments:

Reviewer's Responses to Questions

**Comments to the Author**

1. Is the manuscript technically sound, and do the data support the conclusions?

Reviewer #1: Partly

Reviewer #2: Partly

2. Has the statistical analysis been performed appropriately and rigorously? 

Reviewer #1: Yes

Reviewer #2: Yes

3. Have the authors made all data underlying the findings in their manuscript fully available?

Reviewer #1: Yes

Reviewer #2: Yes

4. Is the manuscript presented in an intelligible fashion and written in standard English?

Reviewer #1: Yes

Reviewer #2: Yes

5. Review Comments to the Author

Reviewer #1: The manuscript SEROLOGICAL TESTING OF BLOOD DONORS TO CHARACTERISE THE IMPACT OF COVID-19 IN MELBOURNE, AUSTRALIA, 2020, with good sampling strategy, estimated SARS-Cov-2 specific antibody prevalence among blood donors in metropolitan Melbourne following a COVID-19 outbreak in the city during summer of 2020. This study provides a surveillance measurement to monitor seroprevalence and infection rates , track outbreaks within communities. As mentioned in the DIDCUSSION, This study has some limitation:

1. Blood donors represents a population less likely to be infected by SARS-CoV-2 than general population;

2. They are normally in better health status to pass the recruitment criteria;

Therefore the test results mostly represent asymptomatic cases. and

3. Antibodies from asymptomatic individuals are usually weaker., which requires higher sensitivity of the assay.

4. Blood donor population doesn't represent people under 16 years, and more and more cases are reported from this age group generally.

Overall, This is a good research study, which reflects the pattern and trends of SARS-Cov-2 infection in metropolitan Melbourne during the outbreak in summer of 2020. Due to the limitation of the selected population, the extent of infection spread and proportion to reported cases of infection could be underestimated. Hope authors can revise their study aim statements.

Reviewer #2: This is an interesting paper looking at serological testing of blood donors to characterize the prevalence of covid-19 from another perspective. Prevalence of infection is related to different status and demographic groups. However, a few remarks:

- In Fig 1: before starting vaccination program

- Results: (Paragraph 1) What is the p value?

- Results: (Paragraph 2) Please show the results in table.

- Figure 2 is not clear.

- Please consider the conclusion of study.

6. PLOS authors have the option to publish the peer review history of their article (what does this mean?). If published, this will include your full peer review and any attached files.

Reviewer #1: No

Reviewer #2: No

---

## [Author Response · Author response to Decision Letter 0]

31 May 2022

Journal Requirements:

Response: The manuscript has been formatted to the journal style requirements.

Response: The ethics statement containing all relevant information has been included. 

Response: The information has now been updated. The study was funded by two organisations: The Victorian Government Department of Health (no grant number) and Snow Medical Foundation (CT28701/G207593).

“The study was supported by the Victorian Government Department of Health and Snow Medical Foundation (CT28701/G207593). The Australian government funds Australian Red Cross Lifeblood to provide blood, blood products and services to the Australian community.”

“The funders have no role in study design, data collection and analysis, decision to publish, or preparation of the manuscript.”

Response: Many thanks, the information has been provided in the cover letter. 

Response: A minimum underlying data set has been uploaded with the re-submission.

Response: The references list has been checked and updated. 

 

Review Comments to the Author

Reviewer #1: The manuscript SEROLOGICAL TESTING OF BLOOD DONORS TO CHARACTERISE THE IMPACT OF COVID-19 IN MELBOURNE, AUSTRALIA, 2020, with good sampling strategy, estimated SARS-Cov-2 specific antibody prevalence among blood donors in metropolitan Melbourne following a COVID-19 outbreak in the city during summer of 2020. This study provides a surveillance measurement to monitor seroprevalence and infection rates , track outbreaks within communities. As mentioned in the DIDCUSSION, This study has some limitation:

1. Blood donors represents a population less likely to be infected by SARS-CoV-2 than general population;

2. They are normally in better health status to pass the recruitment criteria;

Therefore the test results mostly represent asymptomatic cases. and

3. Antibodies from asymptomatic individuals are usually weaker., which requires higher sensitivity of the assay.

4. Blood donor population doesn't represent people under 16 years, and more and more cases are reported from this age group generally.

Overall, This is a good research study, which reflects the pattern and trends of SARS-Cov-2 infection in metropolitan Melbourne during the outbreak in summer of 2020. Due to the limitation of the selected population, the extent of infection spread and proportion to reported cases of infection could be underestimated. Hope authors can revise their study aim statements. 

Response: We have made small changes to clarify that the aim of the study was to measure seroprevalence among blood donors to compare with notifications in Melbourne

Reviewer #2: This is an interesting paper looking at serological testing of blood donors to characterize the prevalence of covid-19 from another perspective. Prevalence of infection is related to different status and demographic groups. However, a few remarks:

- In Fig 1: before starting vaccination program

Response: Paragraph 1 of the materials and methods (last sentence), has been updated with the above detail.

- Results: (Paragraph 1) What is the p value?

Response: The aim of the study was to explore the association between seroprevalence and case notifications and not to formally compare seroprevalence between the three sampling groups which were utilised to inform the sampling method. As such, paragraph 1 which describes the cohort characteristics presented in Table 1, should be interpreted from this perspective. 

- Results: (Paragraph 2) Please show the results in table.

Response: The results described in paragraph 2 were presented in Supplementary Table 2. This table has been moved to the main body of the manuscript. 

- Figure 2 is not clear.

Response: New versions of the figures have been uploaded with the submission. 

- Please consider the conclusion of study.

Response: We agree with the reviewer that the conclusion was somewhat disjointed. We have made some changes in the final discussion paragraphs to better integrate the concluding statements with the study findings.

---

## [Decision Letter · Decision Letter 1]

22 Jun 2022

Serological testing of blood donors to characterise the impact of COVID-19 in Melbourne, Australia, 2020

PONE-D-22-06787R1

Dear Dr. Dorothy Machalek, 

We’re pleased to inform you that your manuscript has been judged scientifically suitable for publication and will be formally accepted for publication once it meets all outstanding technical requirements.

Kind regards,

Shawky M Aboelhadid, PhD

Academic Editor

PLOS ONE

Additional Editor Comments (optional):

Reviewers' comments:

Reviewer's Responses to Questions

**Comments to the Author**

1. If the authors have adequately addressed your comments raised in a previous round of review and you feel that this manuscript is now acceptable for publication, you may indicate that here to bypass the “Comments to the Author” section, enter your conflict of interest statement in the “Confidential to Editor” section, and submit your "Accept" recommendation.

Reviewer #1: All comments have been addressed

2. Is the manuscript technically sound, and do the data support the conclusions?

Reviewer #1: (No Response)

3. Has the statistical analysis been performed appropriately and rigorously? 

Reviewer #1: (No Response)

4. Have the authors made all data underlying the findings in their manuscript fully available?

Reviewer #1: (No Response)

5. Is the manuscript presented in an intelligible fashion and written in standard English?

Reviewer #1: (No Response)

6. Review Comments to the Author

Reviewer #1: (No Response)

7. PLOS authors have the option to publish the peer review history of their article (what does this mean?). If published, this will include your full peer review and any attached files.

Reviewer #1: No

---

## [Editor Report · Acceptance letter]

27 Jun 2022

PONE-D-22-06787R1 

Serological testing of blood donors to characterise the impact of COVID-19 in Melbourne, Australia, 2020 

Dear Dr. Machalek:

I'm pleased to inform you that your manuscript has been deemed suitable for publication in PLOS ONE. Congratulations! Your manuscript is now with our production department. 

Kind regards, 

on behalf of

Professor Shawky M Aboelhadid 

Academic Editor

PLOS ONE